# Genomics of Breast Cancer Brain Metastases: A Meta-Analysis and Therapeutic Implications

**DOI:** 10.3390/cancers15061728

**Published:** 2023-03-12

**Authors:** Thuy Thi Nguyen, Diaddin Hamdan, Eurydice Angeli, Jean-Paul Feugeas, Quang Van Le, Frédéric Pamoukdjian, Guilhem Bousquet

**Affiliations:** 1National Cancer Hospital, Ha Noi 100000, Vietnam; 2Institut National de la Santé Et de la Recherche Médicale (INSERM), Université Paris Cité, UMR_S942 MASCOT, 75006 Paris, Francefrederic.pamoukdjian@aphp.fr (F.P.); 3Department of Pediatrics, Hanoi Medical University, Ha Noi 100000, Vietnam; 4Institut Galilée, Université Sorbonne Paris Nord, 93439 Villetaneuse, France; 5Hôpital La Porte Verte, 78000 Versailles, France; 6Service d’Oncologie Médicale, Hôpital Avicenne, Assistance Publique Hôpitaux de Paris, 93000 Bobigny, France; 7INSERM U1098, 25030 Besançon, France; 8Laboratoire de Biochimie Hôpital Jean Minjoz, Université de Franche-Comté, 25000 Besançon, France; 9Service de Médecine Gériatrique, Hôpital Avicenne, Assistance Publique Hôpitaux de Paris, 93000 Bobigny, France

**Keywords:** breast cancer, brain metastases, genomics, specific gene panel

## Abstract

**Simple Summary:**

Breast cancer brain metastases are a challenging daily practice, and the biological link between gene mutations and metastatic spread to the brain remains to be determined. We performed a meta-analysis on genomic data obtained from primary tumors, extracerebral metastases and brain metastases, to identify gene alterations associated with brain metastatic processes. Fifty-seven publications were selected for this meta-analysis, including 37,218 patients in all, 11,906 primary tumor samples, 5541 extracerebral metastasis samples, and 1485 brain metastasis samples. Using a threshold of 1% for mutation prevalence in the primary tumor, we identified 53 genes, among which 21 were associated with significant differences in prevalence between subgroups (primary tumor, extracerebral metastases, and brain metastases). In particular, we identified six genes with a higher mutation prevalence in brain metastases than in extracerebral metastases: *ESR1, ERBB2, EGFR, PTEN, BRCA2* and *NOTCH1*. These mutated genes could be responsible for the crossing of the blood–brain barrier by cancer cells, and thus have considerable potential therapeutic implications, underlining the added value of obtaining biopsies from brain metastases to develop personalized treatments.

**Abstract:**

Breast cancer brain metastases are a challenging daily practice, and the biological link between gene mutations and metastatic spread to the brain remains to be determined. Here, we performed a meta-analysis on genomic data obtained from primary tumors, extracerebral metastases and brain metastases, to identify gene alterations associated with metastatic processes in the brain. Articles with relevant findings were selected using Medline via PubMed, from January 1999 up to February 2022. A critical review was conducted according to the Preferred Reporting Items for Systematic Review and Meta-analysis statement (PRISMA). Fifty-seven publications were selected for this meta-analysis, including 37,218 patients in all, 11,906 primary tumor samples, 5541 extracerebral metastasis samples, and 1485 brain metastasis samples. We report the overall and sub-group prevalence of gene mutations, including comparisons between primary tumors, extracerebral metastases and brain metastases. In particular, we identified six genes with a higher mutation prevalence in brain metastases than in extracerebral metastases, with a potential role in metastatic processes in the brain: *ESR1, ERBB2, EGFR, PTEN, BRCA2* and *NOTCH1*. We discuss here the therapeutic implications. Our results underline the added value of obtaining biopsies from brain metastases to fully explore their biology, in order to develop personalized treatments.

## 1. Introduction

Brain metastases are becoming a leading cause of mortality among patients with metastatic cancers, including breast cancer. In the last twenty years, the incidence of brain metastases has increased [1,2] as a result of the improved control of metastases outside the central nervous system, and of better imaging techniques that led to improved detection of brain metastases. Despite therapeutic advances, breast cancer brain metastases develop in 15% to 25% of patients [3], mainly due to HER2 overexpressing and triple negative subtypes where they occur in up to 50% of cases [4]. They have a poor life quality impact (neurological deficit, major asthenia, headaches, epileptic manifestations), and a poor prognosis, with a median survival of 16 months [5].

Brain metastatic colonization is a multistep complex process. To migrate into the brain, tumor cells have first to adhere to the brain endothelium, then disrupt the blood–brain barrier, and finally invade through the endothelial layer. Then, only a few tumor cells, which adapt to brain environment, probably have the ability to survive and form brain metastases [6]. Overall, the biological and genomic characteristics of breast cancer cells leading to this process are not fully deciphered [7].

The treatment of brain metastases remains a major therapeutic challenge, with limited indications for and benefits of curative surgery and radiation therapy [8]. Systemic drugs have limited effects on brain metastases, because most anti-cancer drugs fail to cross the blood–brain barrier [9]. Innovative approaches using physical methods or physiological transporters are being explored to facilitate drug penetration across this barrier to deliver them to brain metastases at relevant pharmacological concentrations [9]. New generation tyrosine kinase inhibitors such as osimertinib and tucatinib have been engineered to cross the blood–brain barrier more efficiently, with greater, but time-limited, control of brain metastases [10,11].

Following recent improvements in the genomic sequencing of malignant tumors, targetable genetic alterations are being increasingly identified. However, most alterations have been identified in primary tumors and not metastases [12]. Additionally, this is particularly true for brain metastases, because of the difficulty accessing these localizations [13]. Although primary tumors and metastases can harbor a common genomic signature, significant discrepancies have also been identified between matched samples from a given patient, since metastases can derive from a minority clone in the primary tumor [14]. In addition, there are few studies comparing genomic data from primary tumors and metastases.

Here, we performed the first meta-analysis on genomic data from breast cancer brain metastases. The integrative study we performed enabled us to identify gene alterations associated with brain metastasis, a requisite for the development of new targeted therapies for these localizations.

## 2. Methods

### 2.1. Search Strategy and Selection Criteria

We conducted this systematic review following the methods outlined by Preferred Reporting Items for Systematic Review and Meta-Analyses (PRISMA) [15].

### 2.2. Eligibility Criteria

#### 2.2.1. Inclusion Criteria

Our objective was to perform meta-analysis on genomic data obtained from breast cancer brain metastases. The inclusion criteria were: (1) any study assessing the genomics of breast cancer metastases in any localization, and (2) any article in English from 1999 to the present.

#### 2.2.2. Exclusion Criteria

The following exclusion criteria were applied: (1) studies with unusable or unavailable genomic data on metastases; (2) studies limited to genomic data from primary tumors without available genomic data on metastases; (3) genomic data obtained from samples other than tissue samples (e.g., circulating DNA); (4) reviews, meta-analyses, letters to the editor; (5) experimental data and non-human studies, (6) articles without full text available. The PRISMA flow diagram template used in this study is detailed in Figure 1.

#### 2.2.3. Search Strategy

For a systematic meta-analysis, we searched MEDLINE via PubMed and used the following research algorithm: (“Breast Neoplasms”[Mesh] AND “metast*” AND (“Genomics”[Mesh] OR “mutation*”)). In total, 2776 articles were initially identified. We then tested a second algorithm to focus on brain metastases: “Breast” AND “brain” AND “metast*” AND (“Genom*” OR “mutation*” OR “sequenc*”). We obtained 631 publications. We manually searched the reference lists of all included articles to identify any potentially related articles. Zotero software version 5.0.95.1 was used to manage the references and remove any duplicates. In addition, the references contained in the literature searched and relevant reviews were also considered to avoid eligible articles being missed.

#### 2.2.4. Study Selection

Two authors (TTN and GB) independently screened the papers retrieved, initially by title, then by abstract, and finally by full text.

#### 2.2.5. Protocol and Registration

We registered the review in PROSPERO, an international prospective register of systematic reviews. The protocol can be accessed at: https://www.crd.york.ac.uk/PROSPERO (accessed on 10 September 2021)/Registration number: CRD42021272358.

#### 2.2.6. Quality Assessment

To assess the quality of studies, we used the Q-genie tool [16]. It consists of 11 questions addressing the following aspects of the study methodology: rationale for the study, selection and outcomes, comparability of comparison groups, technical and non-technical exposure, bias, sample size and power, analyses, statistical methods, control for confounders, inferences for genetic analyses and inferences from results. Each question was scored from 1 to 7, as follows: “1 (poor)”, “2”, “3 (good)”, “4”, “5 (very good)”, “6” or “7 (excellent)”. For studies with a control group, a total score of ≤ 35 indicates poor quality, a score of 36–45 indicates moderate quality and a score of *>* 45 indicates good quality [16]. In our meta-analysis on genomic data, as the criterion “non-technical exposure” was not applicable, we considered that a total score of ≤28 indicated poor quality, a score between 29 and 38 indicated moderate quality and a score of >38 indicated good quality.

#### 2.2.7. Statistical Analysis

The data were analyzed using R statistical software (version 4.1.0; R Foundation for Statistical Computing, Vienna, Austria; http://www.r-project.org (accessed on 1 May 2022)). On the basis of the articles selected, we performed a meta-analysis (with the package “meta”) to assess the gene mutation prevalence in breast cancers according to the tumor site: primary tumor, extracerebral metastases and brain metastases. We only considered genes associated with a prevalence ≥1% in the primary tumor.

For all studies, we assessed the gene mutation prevalence according to the quality of studies, good or moderate/poor, and the tumoral site: primary tumor, extracerebral metastases and brain metastases for all tumor samples. We also assessed the gene mutation prevalence in matched tumor samples from the same patient between primary tumor, extracerebral and brain metastases; the genomic method (NGS, targeted NGS or other); and the conditions of tissue preservation (fresh frozen or formalin-fixed). For each outcome, to test the sub-group differences, we used the Q-test based on analysis of variance.

Here is a typical R script example of a single proportion using logit transformation and taking only random effect into account: metaprop(event = name.database$variable.number.of.event, n = name.database$variable.total.number.of.observations, studlab = name.database$name.of.studies, subgroup = name.database$variable.subgroup, sm = “PLOGIT”, common = FALSE, random = TRUE).

We performed a pairwise comparison of gene mutation prevalence across tumor sites using z test, as follows: primary tumors vs. extracerebral metastases, primary tumors vs. brain metastases, and extracerebral metastases vs. brain metastases. We then selected all genes associated with significant differences in prevalence between subgroups and retained only those with prevalence in each tumor site (primary, extracerebral and brain metastases), and with greater prevalence in brain metastases than in the primary tumor. Of these selected genes, we performed univariate and multivariate meta-regression to take into account the effect size of tumor site, quality of studies (good vs. moderate or poor), genomic method (NGS, targeted NGS or other) and tissue preservation condition (frozen or formalin-fixed). Univariate variables yielding *p*-values under 0.20 in the univariate analysis were considered for inclusion in the multivariate analysis. Results of the meta-regression were presented as estimates (ß-coefficient) ± standard error.

We assessed the heterogeneity of the study results using the I2 indicator and Cochran’s Q-test [17]. I2 values of 0%, 25%, 50% and 75% were considered to indicate absence, low, moderate and marked heterogeneity, respectively. A *p*-value ≤ 0.05 of the Q tests indicated significant heterogeneity. Due to a significant heterogeneity in gene mutation prevalence, pooled results were summarized using the random-effect model and ordered in decreasing prevalence values (%) with their 95% Confidence Interval (95%CI), including a comparison of subgroup prevalence.

Graphically, the gene mutation prevalence according to the tumor site was presented as a heatmap plot. Gene mutations associated with an increased prevalence in brain metastases were presented as a bar plot of frequencies ordered in decreasing values in primary tumors. Copy-number alterations (LOH, material gain) were also presented as a bar plot of frequencies. All tests were two-tailed, and the threshold for statistical significance was set at a *p*-value ≤ 0.05.

This is a meta-analysis on published data, so ethical approval was not sought.

## 3. Results

### 3.1. Study Selection, Characteristics and Quality Assessment

After the literature search and removal of duplicate articles, we identified 3394 studies. By careful selection based on titles and abstracts, 3337 studies were excluded, mainly because genomic data were not available. A total of 57 studies were finally included in this meta-analysis (Figure 1).

The characteristics of the 57 studies selected are summarized in Appendix A. Our meta-analysis included a total of 37,218 patients with a median age at diagnosis of 53.5 years, and a total of 18,932 samples, including 11,906 primary tumor samples, 5541 extracerebral metastasis samples, and 1485 brain metastasis samples.

For the quality assessment, 29 studies (50.9%) were good quality, 25 studies (43.8%) were of moderate quality, and three studies (5.3%) were poor quality (Appendix A). However, since good quality studies included much larger numbers of tumor samples than other studies, they accounted for 97.3%, 97.5% and 85.3% of the total number of samples for primary tumors, extracerebral and brain metastases, respectively.

### 3.2. Gene Mutation Profiles in Breast Cancer Brain Metastases

Using a threshold of 1% for mutation prevalence in the primary tumor, we identified 53 genes. Considering all pooled tumor samples, we first compared gene mutation prevalence for these 53 genes between studies of good quality and studies of moderate/poor quality. For 16 genes, the mutation prevalence was significantly higher for moderate and poor-quality studies, but with small sample numbers analyzed (Appendix A).

Then, we compared gene mutation prevalence according to tumor sample site: primary tumor, extracerebral metastases, and brain metastases (Figure 2A). Five of them had a mutation prevalence over 10% in all three types of samples: *TP53, PIK3CA, MYC, KMT2C* and *ATRX*. In brain metastases, mutation prevalence was particularly high for *TP53* (58%), *FOXA1* (43%), *FGFR4* (33%), *BRCA2* (22%), *FGFR2* (20%), *BRAF* (19%) and *PTEN* (15%).

Among the 53 genes initially retained, 21 were associated with significant differences in prevalence between subgroups (Appendix A). When we compared brain metastases with extracerebral metastases, the mutation prevalence was significantly higher for 10 genes. We considered that the mutation prevalence was reliable if a minimum sample number of 100 brain metastases was analyzed. We finally retained six genes: *TP53, BRCA2, PTEN, NRAS, NOTCH1* and *EGFR* (Table 1, Figure 2B and Appendix A for *BRCA2, PTEN, NOTCH1* and *EGFR*).

We then analyzed matched tumor samples between primary tumor, extracerebral metastases and brain metastases. Most differences observed did not reach statistical significance due to small sample numbers (Appendix A). When we considered the most frequently mutated genes, such as *TP53, PIK3CA* or *BRCA2*, the gene mutation prevalence was comparable with data obtained from all pooled tumor samples (Appendix A).

### 3.3. Heterogeneity of Gene Mutation Prevalence between Studies Was Less Marked for Brain Metastasis Samples

Among the 53 genes, there was significant heterogeneity across studies except for 26 genes in the brain metastasis samples (Appendix A). To address this limitation, we performed several subgroup analyses. We first assessed the prevalence of gene mutations in all samples according to the genomic analysis method (NGS, targeted NGS and others) and tumor preservation conditions (frozen vs. formalin-fixed). NGS was the main method used, for 61.4% of the studies and 91% of all tumor samples, with persistent significant heterogeneity across studies. For mutation prevalence, a significant difference across genomic analysis methods was only observed for nine genes, of which seven had a higher mutation prevalence with targeted NGS than with whole exome/genome NGS (Appendix A).

For the conditions of tumor preservation, freezing is usually considered to be the standard condition for whole genome analyses [18]. It accounted for 28% of studies and only 7.1% of tumor samples, with less heterogeneity across studies. In addition, mutation prevalence was significantly higher for 16 genes when the frozen condition was compared to the formalin-fixed condition (Appendix A).

### 3.4. Copy-Number Alterations and Loss of Heterozygosity in Breast Cancer Brain Metastases

For gene amplification, data were obtained for 11,950 patients and 9286 samples. When we analyzed and compared the prevalence between primary tumor, extracerebral metastases and brain metastases, most differences observed did not reach statistical significance due to small sample numbers, especially for extracerebral and brain metastases. Interestingly, for *PTEN*, the mutation prevalence decreased from 11% to 0% (Appendix A).

For the loss of heterozygosity and other copy-number alterations, only seven studies with 628 patients and 614 samples provided data. Due to the small number of samples analyzed for each gene and each tumor site, we were not able to perform reliable statistical comparisons (Appendix A). Interestingly, in the three loci 9p21.3, 10q23.31 and 17q11.2, the prevalence of LOH increased between the extracerebral metastases and brain metastases. For example, for the 10q23.31 locus, comprising *PTEN,* the prevalence in brain metastases was 75%, but in only 10 brain samples (Figure 2C).

*ESR1, ERBB2, EGFR, PTEN, BRCA2,* and *NOTCH1* mutations could be linked to metastatic processes in the brain.

For the six genes with a minimum sample number of 100 brain metastases analyzed and a mutation prevalence that was higher in brain metastasis samples than in extracerebral metastases, we ran univariate and multivariate meta-regressions to determine whether gene mutation prevalence was associated with sample subgroups. Focusing on brain metastasis localizations, we showed that they significantly influenced mutation prevalence for four genes: *EGFR, PTEN, BRCA2* and *NOTCH1* (Table 2). We also decided to retain *ESR1* and *ERBB2,* since their mutation prevalence increased gradually from primary tumor to extracerebral metastases and brain metastases (Appendix A, Figure 2B and Figure 3), an observation that may have therapeutic implications. For these six genes, we produced a cartography of the mutations reported in our meta-analysis (Figure 3).

For *ESR1,* all reported mutations were common to extracerebral and brain metastases, located in the ligand binding domain, which includes the transcriptional activating function-2 (AF2). These mutations are activating mutations; they can promote cell growth in the absence of estradiol [19,20,21], and they could be linked to an increased risk of brain metastases, for instance the pD538G mutation [22].

For *ERBB2*, most mutations were common to extracerebral and brain metastases. Most of them are activating mutations, located in the tyrosine kinase domain. pP1227S was the only mutation restricted to a brain metastasis sample. It is located in the autophosphorylation domain, and could be responsible for constitutive receptor activation as described for pY1068F and pY992F *EGFR* mutations [23,24].

For *EGFR*, a pL858R mutation was found in one brain metastatic sample. This hotspot mutation, well-described in non-small-cell lung adenocarcinoma, is predictive of the response to EGFR tyrosine kinase inhibitors [25,26]. pE282K and pT594N mutations are located in the extracellular ligand-binding domain. *EGFR* ectodomain mutants are oncogenic in preclinical models of glioblastoma [27].

For *PTEN*, three mutations were identified in brain metastasis samples: pE18fs, pQ171X and pE242*. Both pE18fs and pQ171X are located in the phosphatase domain, which is important for the membrane-binding of PTEN and facilitates phospholipid hydrolysis [28,29]. Mutations occurring in the phosphatase domain, such as pY65S, pQ171R, pC124S, and pG129E, have been associated with the loss of PTEN function [30,31]. The pE242* mutation is located in the C2 domain, which interacts with the phosphatase domain, regulating PTEN phosphatase activity. It can also bind to phospholipid membranes, enabling PTEN to inhibit cell migration [29,32].

For *BRCA2*, two mutations were identified in brain metastasis samples: pH2563R and pE3343K. The pE3343K mutation is located between the NLS2 and NLS3 regions, and its functional value is as yet undeciphered [33]. The pH2563R mutation is located in the helical domain within the DNA binding domain, which is in turn implicated in BRCA2 binding to both single-strand and double-strand DNA. This mutation could be responsible for abnormal DNA repair, as described for the G2609D mutation [34].

The two mutations of *NOTCH1* identified in brain metastatic samples were pL912W and pH2107D. pL912W is located in the EGF-like domain, involved in ligand binding, and it could be oncogenic [35]. pH2107D is located in the ankyrin domain, which is responsible for mediating protein–protein interactions [36].

Overall, some gene mutations could be linked to the breast cancer brain metastatic process, with strong therapeutic implications.

## 4. Discussion

Here, we report the first meta-analysis of genomic data concerning 37,218 patients with metastatic breast cancers, including 1485 brain metastasis samples. We have recently shown that this methodological approach provides more reliable gene mutation prevalence values than data obtained from individual sources [12]. In addition, the stringent methodology we have used is a strength of our study, with two complementary search algorithms, careful selection of the studies, quality control of the studies, and an approach to explaining heterogeneity across subgroups. In a recent review on genomic data for 164 breast cancer brain metastases, the gene mutation prevalence for the two most frequently mutated genes, *TP53* and *PIK3CA,* was similar to those in our meta-analysis [37]. In contrast, there were many discrepancies for the other genes. In that review, the mutation prevalence was 4% for *PTEN* and *BRCA2*, compared to 15% and 22%, respectively, in our meta-analysis, in which a much larger number of brain metastasis samples were analyzed.

Our meta-analysis highlights the need to sequence brain metastases, and thus to obtain tissue samples, which could be facilitated by the use of imagery-guided biopsies [13]. Our meta-analysis has also shown the added value of using targeted NGS to provide reliable data on gene mutation prevalence. Compared with whole exome/genome sequencing, targeted NGS is faster and less costly, with a greater sensitivity to detect mutations with low allelic frequency [38]. This type of approach could be proposed to sequence a panel of genes with therapeutic implications on brain metastasis samples.

One value of our meta-analysis is that we compared genomic data obtained from brain metastases, extracerebral metastases and primary tumors. Metastatic cells can derive from minority clones within a primary tumor [14], but also from minority clones within extracerebral metastases, with mutations linked to site-specific metastatic spread [39,40,41]. Indeed, a brain metastasis is the result of biological properties favoring the crossing of the blood–brain barrier by cancer cells, and their implantation in the brain. Biological factors associated with increased risk of brain metastases are not fully understood. There are transcriptomic signatures of primary tumors or extracerebral metastases that are associated with an increased risk of brain metastases [42,43,44,45]. However, there are no data demonstrating that some gene mutations could be responsible for the crossing of the blood–brain barrier by cancer cells.

In our meta-analysis, we identified six genes with high mutation prevalence in brain metastases, of particular interest for their potential role in brain metastatic process and resistance to first-line anti-cancer drugs: *ESR1, ERBB2, EGFR, PTEN, BRCA2* and *NOTCH1* (Figure 4).

*ESR1* encodes for the estrogen receptor 1 protein. After estrogen binding, ESR1 translocates to the nucleus and binds to estrogen receptor elements in enhancer regions of the genome, mediating gene transcription during normal physiological processes, but also in the course of breast cancer tumorigenesis [46]. Activating mutations in the ligand-binding domain of *ESR1* have been observed in 10% to 40% of metastatic ER+ breast cancers, conferring endocrine resistance [46,47].

*ERBB2* is a proto-oncogene encoding a member of the epidermal growth factor receptor family. *ERBB2* amplification is common in different cancer types [48,49]. It is a well-established risk factor for brain metastases in breast cancer. However, the biological mechanisms for this association are not fully deciphered. In vitro, HER2 overexpressed cancer cells are more likely to interact with integrin β4, promoting their adhesion to endothelial cells of the blood–brain barrier [50]. Additionally, in a murine model of brain metastases, HER2 overexpression increases the outgrowth of metastatic tumor cells in the brain [51]. ERBB2 mutations are less common, with prevalence ranging from 0.2% to 12.6% [52,53]. In breast cancer, *ERBB2* mutations have been described in all histological subtypes, usually in the absence of *ERBB2* amplification [54,55]. In preclinical studies, *ERBB2* mutations located in the extracellular and the C-terminal domains are usually predictive of sensitivity to trastuzumab, whereas most mutations in the tyrosine kinase domain are resistant mutations. For example, pL755P/S mutations, common in breast cancer, are associated with resistance to lapatinib, an anti-HER2 TKI. In contrast, the A775_776insYVMA mutation, frequently identified in lung cancer brain metastases, is associated with response to afatinib and neratinib [52,56].

*EGFR* is a frequently altered oncogene. *EGFR* activation, through either amplification or mutation, in turn activates numerous downstream signal transduction pathways including the Ras-Raf-MAPK and PI3K/Akt pathways [57]. In preclinical models, the association between EGFR pathway activation and breast cancer brain metastases, including through EGFR ligand heparin binding EGF (HBEGF) overexpression, was also reported [7,58]. *EGFR* mutations, usually ligand-independent activating mutations, are well-known oncogenic events in non-small-cell lung cancers [40,59]. In addition, in 384 patients with non-small-cell lung cancer, the incidence of brain metastases was 49.5% among patients with L858R *EGFR*-mutated cancer versus 27.3% among those with wild-type cancer [60].

*PTEN* is a tumor suppressor gene. The PTEN protein is mainly involved in the blockade of PI3K/Akt signaling originating from EGFR activation. Inactivating *PTEN* mutations have been identified in many cancer types, particularly endometrial carcinomas and glioblastomas [61,62]. In 56 brain metastases from different cancer types, the prevalence of *PTEN* loss was very high in cases of lung and breast cancers, sometimes combining LOH and an inactivating mutation, suggesting that PTEN loss of function could contribute to brain metastatic processes [63]. The loss of PTEN was also found to predict trastuzumab resistance among breast cancer patients [64].

*BRCA2* is a tumor suppressor gene. The BRCA2 protein plays an important role in DNA repair and transcription regulation. *BRCA2* germline mutations are associated with an increased risk of breast, ovarian, and pancreas cancers [65,66]. In breast cancer, *BRCA2* germline mutations have been found to be significantly associated with brain metastasis, regardless of tumor subtype [67].

*NOTCH1* encodes a trans-membrane receptor that belongs to a well-conserved signaling pathway. When NOTCH1 is activated, it splits to form an extracellular domain and an intracellular domain, itself translocated to the nucleus to regulate the transcription of target genes. Activating *NOTCH1* mutations have been identified in different cancer types [68]. In preclinical models of breast cancer, NOTCH1 signaling pathway activation has been associated with an increased risk of brain metastases [69,70].

These genes also have considerable potential therapeutic implications.

For *ESR1* mutations, estrogen receptor antagonists such as fulvestrant appear broadly effective in vitro, in particular the D538G mutant [20]. In mice, the combination of fulvestrant and palbociclib or everolimus inhibits tumor growth in breast cancers harboring D538G or Y537S *ESR1* mutations [71]. In patients with endocrine-resistant breast cancer, the same combinations were efficacious [72]. Other ESR1 targets, such as lasofoxifene and H3B-5942, have proved superior to fulvestrant in inhibiting metastatic processes in breast cancer xenografts harboring Y537S and D538G *ESR1* mutants [73,74]. Their benefits for the treatment of women with ER-positive breast cancer with acquired *ESR1* mutations are currently being assessed in clinical trials [75,76].

In *ERBB2*-mutated cancers, the benefit of anti-HER2 TKIs has been evaluated. In a phase II trial involving 125 patients with *ERBB2* mutations across 21 cancer types, treatment with neratinib, a pan HER2-TKI, provided a 24% response rate among breast cancer patients with *ERBB2* S310, L755, V777, G778_P780dup and Y772_A775dup mutations [77]. Among 16 patients with *ERBB2*-mutated cervical cancers, the response to neratinib was linked to the pS310F mutation [78].

Typically, *EGFR* mutations are associated with high response rates to anti-EGFR TKIs in metastatic non-small-cell lung cancer. This is also true for brain localizations, with response rates ranging from 36.5% to 91% [79,80]. In particular, osimertinib, a third-generation anti-EGFR, has better brain penetration, with response rates of over 70% [80]. In a preclinical study, osimertinib also showed marked efficacy in *EGFR*-mutated glioblastoma [81]. For a metastatic breast cancer patient with *EGFR* L861Q mutation in a resort situation, treatment with anti-EGFR provided 6 months of disease control [82].

The PI3K/AKT/mTOR pathway is frequently activated in breast cancer brain metastases due to PTEN loss of function and frequent *PIK3CA, AKT* and *mTOR* activating mutation, as evidenced in our meta-analysis. In a murine model of brain xenografts derived from HER2-overexpressed breast cancer with PTEN loss, a combination of PI3K and mTOR inhibitors considerably inhibited tumor growth [83]. Since a loss of PTEN decreases homologous recombination and sensitizes tumor cells to polyadenosine diphosphate ribose polymerase (PARP) inhibitors, a combination of PARP-inhibitor with PIK3-inhibitor could be promising for cancers with *PTEN* loss of function mutations [84].

PARP inhibitors are currently approved for the treatment of several metastatic cancers with *BRCA* mutations [85,86,87]. Since brain metastases occur in approximately half the patients with advanced breast cancer with *BRCA* mutations [67], and *BRCA2* mutation prevalence reached 22% in our meta-analysis, PARP inhibitors could be promising tools for the treatment of brain metastases. Indeed, for a woman with brain metastases of endometrial cancer origin and with a circulating *BRCA1* mutation, treatment with a PARP inhibitor provided an excellent response in brain localizations [88]. In preclinical models of triple-negative breast cancer brain metastases, carboplatin in combination with veliparib, a PARP inhibitor, decreased tumor volume in the *BRCA*-mutant [89]. An clinical trial is testing the combination of cisplatin and veliparib for breast cancer brain metastases harboring *BRCA* mutations [90].

Finally, the NOTCH pathway is frequently activated in metastatic cancers, leading to the development of NOTCH-targeted therapies [91,92]. In preclinical models of breast cancer, γ-secretase inhibitors showed promising activity in brain metastases [69,70].

To date, several commercialized drugs, such as osimertinib or neratinib, are not currently approved for the treatment of metastatic breast cancer. Their use shall be discussed during dedicated multidisciplinary meetings. Additional clinical trials are required, ideally basket trials dedicated to brain metastases of different cancer types.

Our meta-analysis has some limitations. First, it was performed on aggregated data and not individual data. For this reason, some subgroup analyses could not be assessed because of missing data (data on ethnicity, histological subtypes). Second, there was significant heterogeneity across studies for mutation prevalence, usually persisting despite various subgroup analyses to address this limitation. Particularly, since heterogeneity in genomic studies could be linked to tumor heterogeneity itself, we intended to assess gene mutation prevalence, comparing multiple sampling with single sampling. However, data were only available for 7% of the studies, accounting for 0.5% of the total number of samples analyzed. On the other hand, this heterogeneity disappeared for most genes when we considered solely brain metastasis samples. According to the seed and soil hypothesis, brain metastases can derive from a minority clone within a primary tumor or from another metastatic localization [14,93] with possible specific signatures linked to organ-specific metastatic sites [94]. This highlights the need for biopsy and brain metastasis analyses.

## 5. Conclusions

This is the first meta-analysis of genomic alterations in breast cancer brain metastases. Our results underline the added value of obtaining biopsies from brain metastases to fully explore their biology, for the development of personalized treatments.

## Figures and Tables

**Figure 1 cancers-15-01728-f001:**
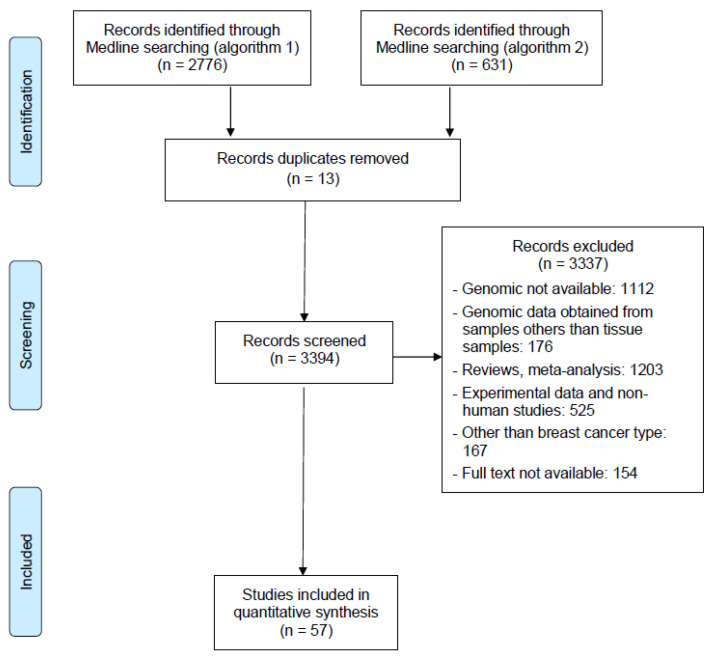
PRISMA flowchart for the screening and selection of the studies.

**Figure 2 cancers-15-01728-f002:**
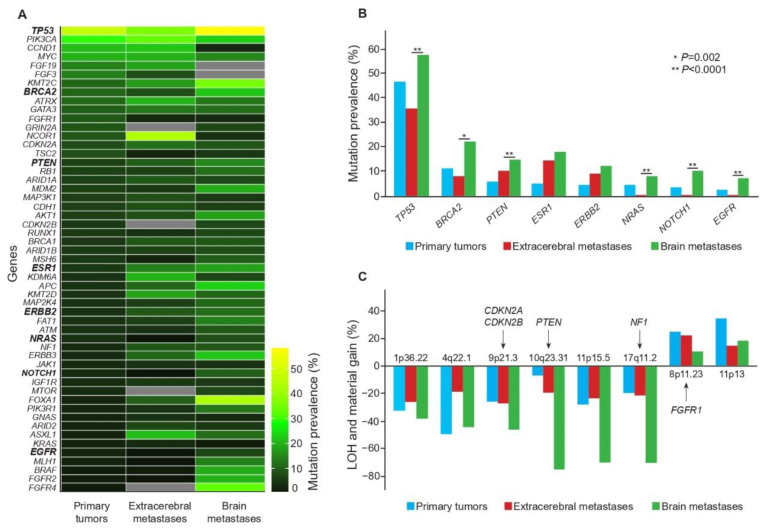
Gene mutation prevalence, loss of heterozygosity and copy number alterations in primary tumor, extracerebral metastases and brain metastases. Panel (**A**) shows a heatmap plot for the prevalence of gene mutations according to the tumor site (primary tumor, extracerebral metastases and brain metastases). Panel (**B**) shows the increased prevalence of gene mutations in brain metastases for 8 genes: *TP53, BRCA2, PTEN, ESR1, ERBB2, NRAS, NOTCH1, EGFR*. *p*-values reported here corresponded to those of Table 1 and of the pairwise comparison of gene mutation prevalence between brain metastases and extracerebral metastases. Panel (**C**) shows the loss of heterozygosity and material gain identified in breast cancer brain metastases.

**Figure 3 cancers-15-01728-f003:**
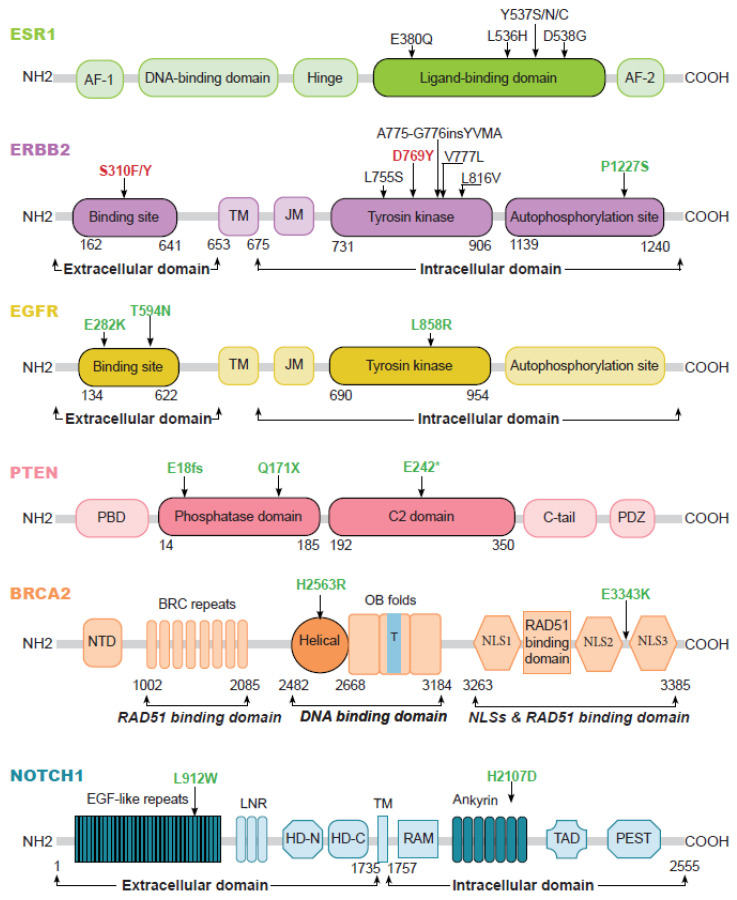
Cartography of the mutations reported in the meta-analysis for the 6 genes with an increased mutation prevalence in brain metastases. Mutations exclusive to brain metastases are identified in green. Those exclusive to extracerebral metastases are in red, and the mutations common to both sites are in black. AF-1: activation function-1, AF-2: activation function-2, TM: transmembrane, JM: juxtamembrane, PBD: PIP2-binding domain; NTD: N-terminal domain, OB folds: oligonucleotide binding folds, T: tower domain, NLS: nuclear localization sequence, EGF-like repeats: epidermal growth factor-like repeats, LNR: Lin12/Notch repeat, HD-N: heterodimerization domain- N terminal, HD-C: heterodimerization domain- C terminal, RAM: Rbp-associated molecule, TAD: transactivation domain, PEST: a region rich in prolone (P), glutamate (E), serine (S) and threonine (T), *: means there is a codon stop.

**Figure 4 cancers-15-01728-f004:**
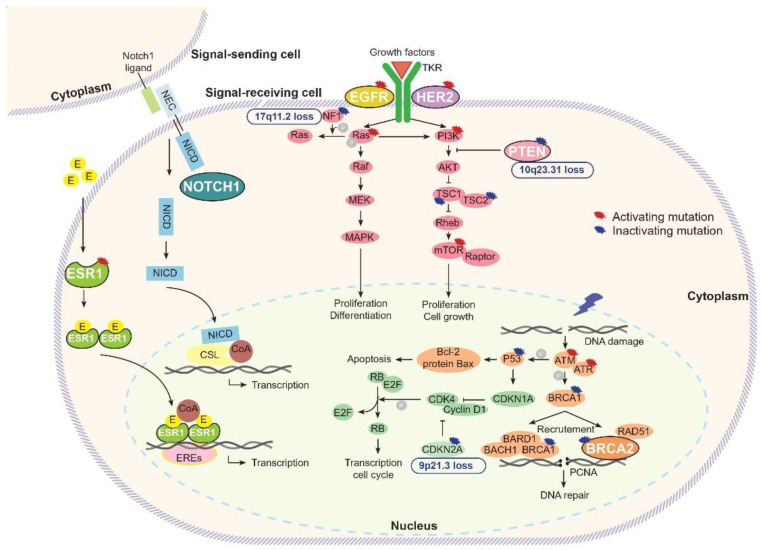
Gene alterations and signaling pathways involved in breast cancer carcinogenesis and brain metastatic processes: estrogen, CoA: coactivators, EREs: estrogen receptor elements, NEC: Notch1 extracellular cell, NICD: Notch1 intracellular cell domain, CSL (CSL proteins): CBF-1/RBPJ-κ in Homo sapiens/Mus musculus, respectively, Suppressor of Hairless in Drosophila melanogaster, Lag-1 in Caenorhabditis elegans, PCNA: proliferating cell nuclear antigen.

**Table 1 cancers-15-01728-t001:** Pairwise comparison of prevalence of gene mutations according to the tumor site.

	*p*-Value for Pairwise Comparisons of Mutation Prevalence		
Gene	Primary Tumors vs. Extracerebral Metastases	Primary Tumors vs. Brain Metastases	Extracerebral Metastases vs. Brain Metastases
** *TP53* **	<0.0001	<0.0001	**<0.0001**
*PIK3CA*	0.06	0.04	0.33
** *BRCA2* **	0.002	<0.0001	**0.002**
** *PTEN* **	0.008	<0.0001	**<0.0001**
*CDKN2B*	NA	0.95	NA
*BRCA1*	<0.0001	0.002	0.72
*KDM6A*	0.06	0.02	0.34
** *NRAS* **	<0.0001	0.35	**<0.0001**
*NF1*	<0.0001	<0.0001	0.8
*ERBB3*	0.65	<0.0001	0.8
** *NOTCH1* **	<0.0001	<0.0001	**<0.0001**
*MTOR*	NA	0.001	NA
** *FOXA1* **	<0.0001	<0.0001	**<0.0001**
*PIK3R1*	0.68	<0.0001	0.06
*ARID2*	0.02	0.004	0.77
*ASXL1*	<0.0001	<0.0001	0.05
** *EGFR* **	<0.0001	<0.0001	**<0.0001**
** *MLH1* **	0.001	<0.0001	**<0.0001**
** *BRAF* **	0.0001	<0.0001	**<0.0001**
** *FGFR2* **	0.29	<0.0001	**<0.0001**
*FGFR4*	NA	<0.0001	NA

Bold = significant *p*-value at the threshold of 0.05, NA: not available.

**Table 2 cancers-15-01728-t002:** Univariate and multivariate meta-regression.

Gene	Samples	Univariate Meta-Regression			Multivariate Meta-Regression		
		Estimate	Standard Error	*p*	Estimate	Standard Error	*p*
*TP53*	**Tumor site:** Primary tumors	1 (ref)	-				
	Extracerebral metastases	−0.42	0.27	0.11			
	Brain metastases	0.44	0.27	0.1			
	**Quality of studies** (good)	−0.29	0.24	0.22			
	**Method analysis:** NGS	1 (ref)	-	-			
	Targeted NGS	0.24	0.25	0.33			
	Other	−0.5	0.47	0.28			
	**Preservation** (Frozen)	−0.25	0.29	0.39			
** *BRCA2* **	**Tumor site:** Primary tumors	1 (ref)	-	-	1 (ref)	-	-
	Extracerebral metastases	−0.24	0.35	0.49	−0.12	0.41	0.77
	Brain metastases	0.94	0.35	**0.008**	0.92	0.37	**0.01**
	**Quality of studies** (good)	−0.79	0.44	0.07			
	**Method analysis:** NGS	1 (ref)	-	-			
	Targeted NGS	−0.29	0.4	0.48			
	Other	-	-	-			
	**Preservation** (Frozen)	−0.87	0.66	0.19	−0.59	0.58	0.31
** *PTEN* **	**Tumor site:** Primary tumors	1 (ref)	-	-	1 (ref)	-	-
	Extracerebral metastases	0.46	0.32	0.15	0.47	0.29	0.1
	Brain metastases	0.84	0.35	**0.01**	0.32	0.35	0.36
	**Quality of studies** (good)	−1.26	0.32	**<0.0001**	−1.21	0.37	**0.01**
	**Method analysis:** NGS	1 (ref)	-	-			
	Targeted NGS	−0.41	0.38	0.27			
	Other	−0.58	0.78	0.45			
	**Preservation** (Frozen)	0.88	0.37	**0.01**	0.23	0.36	0.52
*NRAS*	**Tumor site:** Primary tumors	1 (ref)	-	-	1 (ref)	-	-
	Extracerebral metastases	−3.81	1.12	**0.0007**	−3.05	0.88	**0.0005**
	Brain metastases	0.88	0.89	0.32	0.65	0.71	0.35
	**Quality of studies** (good)	−3.42	1.11	**0.002**	−1.95	0.64	**0.002**
	**Method analysis:** NGS	1 (ref)	-	-			
	Targeted NGS	-	-	-			
	Other	−1.62	2.38	0.49			
	**Preservation** (Frozen)	-	-	-			
** *NOTCH1* **	**Tumor site:** Primary tumors	1 (ref)	-	-	1 (ref)	-	-
	Extracerebral metastases	−3.55	1.18	**0.002**	−2.97	1.1	**0.006**
	Brain metastases	1.46	0.87	0.09	1.5	0.76	**0.04**
	**Quality of studies** (good)	−1.52	1.09	0.16	−0.39	0.81	0.62
	**Method analysis:** NGS	1 (ref)	-	-	1 (ref)	-	-
	Targeted NGS	1.8	0.99	0.07	1.04	0.75	0.16
	Other	-	-	-	-	-	-
	**Preservation** (Frozen)	1.12	1.21	0.35			
** *EGFR* **	**Tumor site:** Primary tumors	1 (ref)	-	-	1 (ref)	-	-
	Extracerebral metastases	−1.89	0.78	**0.01**	−1.42	0.39	**0.0003**
	Brain metastases	1.38	0.61	**0.02**	1.46	0.44	**0.001**
	**Quality of studies** (good)	−1.83	0.77	**0.01**	−1.49	0.46	**0.001**
	**Method analysis:** NGS	1 (ref)	-	-			
	Targeted NGS	0.33	0.79	0.67			
	Other	-	-	-			
	**Preservation** (Frozen)	−0.99	1.67	0.55			

Uni- and multivariate meta-regressions were run to assess sample groups significantly associated with prevalence of gene mutations. Sample groups yielding *p*-values under 0.20 in the univariate analysis were considered for inclusion in the multivariate analysis. Bold = significant *p*-value at the threshold of 0.05.

## Data Availability

Data collected for this study are readily available, as all included articles in this meta-analysis are publicly accessible through the PubMed Library.

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
