# Peer review of "Genomics of Breast Cancer Brain Metastases: A Meta-Analysis and Therapeutic Implications"

_cancers, 2023, doi:10.3390/cancers15061728_

Round 1

Reviewer 1 Report

Overall this was an informative review that needed some language edits. I have attached some typo-fixing edits.

Author Response

We thank Reviewer#1 for these comments. Following the advice of Reviewer#1, we did the modification accordingly.

Reviewer 2 Report

Breast cancer brain metastases are challenging daily practice, and the biological link between gene mutations and metastatic spread to the brain remains to be determined. In this study they performed a meta-analysis on genomic data obtained from primary tumors, extracerebral metastases and brain metastases, to identify gene alterations associated with brain metastatic processes.  They identified 6 genes with a higher mutation prevalence in brain metastases than in extracerebral metastases: ESR1, ERBB2, EGFR, PTEN, BRCA2 and NOTCH1. It is suggested that these genes may have potential therapeutic implications, although the clinical applications of this are unclear.

Author Response

In the discussion section of our manuscript, we had written a paragraph on therapeutic applications for each of the 6 genes. However, we agree that many available drugs such as osimertinib or neratinib are not currently approved for the treatment of metastatic breast cancer. Clinical trials are required, ideally basket trials dedicated to brain metastases of any cancer type.

To follow the remark by Reviewer#2, we have added a sentence in the discussion section of the revised version of our manuscript, lines 464 to 467 “To date, several commercialized drugs such as osimertinib or neratinib are not currently approved for the treatment of metastatic breast cancer. Their use shall be discussed during dedicated multidisciplinary meetings. Additional clinical trials are required, ideally basket trials dedicated to brain metastases of different cancer types”.

Reviewer 3 Report

Authors present here a very interesting meta-analysis to fill a gap in the knowledge about the breast cancer brain metastasis. They perfectly respect the guidelines to prepare their study but they have to go further in the pathophysiological implications of their conclusions. I thus suggest some revisions before publication. The different comments are presented after.

Line 17: I think that is an error in the name of the corresponding author. Prof. Guilhem has to change in Prof. Bousquet

Line 23: a space is lacking before Fifty

Lines 25-27: authors have to explain the subgroups

Lines 28-30: I agree about the therapeutic implication but I think that is important to link the results to the physiopathological mechanisms.

Lines 35-38: I think that the algorithms are not necessary to the abstract

Line 46: a space is lacking before Our

Introduction: this section has to be expanded. Indeed, the  authors have to add information about all the metastatic profile, life quality impact of brain metastasis. They should also explain the metastatic process.

Line 110: authors have to add reference to support the use of the Q-genie Tool.

Lines 152-153: same thing about the I2 indicator and the following test.

Lines 164-165: the last sentence should be presented in a separate section about ethic statement.

Lines 174: table S1 must be completed. to make reading easier, it could be interesting to find a summary of the clinical data mentioned in the text at the level of the table: age, number of cases...

Lines 192-193: based on the color scale, it seems that KMT2C also shows a high prevalence in the brain metastasis.

Lines 198-200: please add markers on the table 1to find easily the identified genes.

Lines 197-198: what was the basis to select a cut-off at 100 samples?

Lines 213-214: check the accuracy of the legend regarding the panel C.

Figure 3: I’m not sure that is necessary in the main text. I suggest to move it in the supplementary material

Lines 224-225: add a space between the two parts

Lines 241-244: I suggest to move this paragraph in the discussion section when you analyse the limits of the study

Lines 257-258: why the conclusion appears here? Why the list is different to the previous one?

Lines 264-265: authors add a new criteria in their analysis. This one must be presented in the method section

Lines 268-270: this point is very important and I think that it should be more develop in the discussion section. Indeed, it should be very interesting to link the mutation to the cell mechanisms explaining the metastatic process.

Table 2: enlarge the first column to have the genes’ names on one row

Lines 307-308: the additional value of the study is clear. However, few elements are compared between brain metastasis and the other extracerebral metastasis. I think to improve the study you have to deeply analyze these comparisons.  

Figure 6: you have to highlight your genes panel on the figure and you must to increase the size of the legend (activating or inactivating mutations).

Lines 330-340: you have to remember the link between ERBB2 et HER2.

Lines 341-347: discussion about the EGFR is based on the lung cancer. However a large panel of article in breast cancer is also available. Thus, I think that you must use the closest illustration of your work.

Lines 324-366: in this first part of the discussion, the authors present some putative physiopathological links with the genes identified in the study. I suggest to improve the impact of the study to enlarge this section. Indeed, it should be very interesting to link the results with cellular processes involved in metastatic process. To cite studies presenting only a link between expression variations and cancers/metastasis is not enough. Authors have to describe cell mechanisms. In addition, the selection of the articles used in the meta-analysis is well described but some elements of the literature have been omitted. For instance, Massague et al demonstrated that COX2, HBEGF and ST6GALNAC5 are major mediator of the brain targeting (PMID: 19421193).

Line 369: in vitro should be in italic

Lines 394-397: authors have to precise the model allowing the conclusions used here.
